# Energy-Aware Imitation Learning for Steering Prediction Using Events and Frames

## Abstract

In autonomous driving, relying solely on frame-based cameras can lead to inaccuracies caused by factors like long exposure times, high-speed motion, and challenging lighting conditions. To address these issues, we introduce a bio-inspired vision sensor known as the event camera. Unlike conventional cameras, event cameras capture sparse, asynchronous events that provide a complementary modality to mitigate these challenges. In this work, we propose an energy-aware imitation learning framework for steering prediction that leverages both events and frames. Specifically, we design an **E**nergy-driven **C**ross-modality **F**usion **M**odule (ECFM) and an energy-aware decoder to produce reliable and safe predictions. Extensive experiments on two public real-world datasets, DDD20 and DR-Fuser, demonstrate that our method outperforms existing state-of-the-art (SOTA) approaches. The codes will be released upon acceptance.

## 1 Introduction

End-to-end learning has made remarkable strides in steering prediction for autonomous driving by enabling direct mapping from visual observations to motion estimation Bojarski (2016); Xu et al. (2017); Chen et al. (2024); Zhou et al. (2024a). A commonly used sensor for visual observations is the frame-based camera, which has shown promising results in motion estimation when used as a single-modality input Prakash et al. (2020); Ohn-Bar et al. (2020). However, frame-based cameras often experience a substantial performance drop in challenging conditions, such as high-speed motion and adverse lighting (e.g., low-light and overexposure) Gallego et al. (2020); Cannici & Scaramuzza (2024); Aydin et al. (2024); Zubic et al. (2024). An emerging bio-inspired vision sensor, the event camera, such as dynamic and active pixel vision sensors (DAVIS), has gained attention in the field of vision perception Brandli et al. (2014); Moeys et al. (2017). Un-

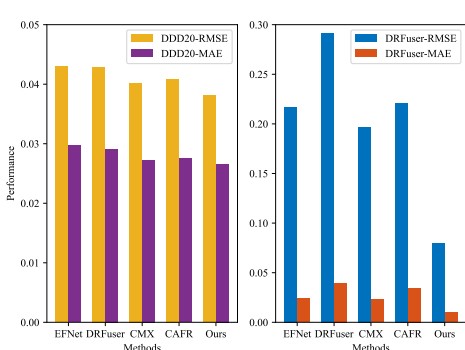

Figure 1: Performance comparison on the DDD20 and DRFuser datasets. Our proposed method achieves SOTA performance, outperforming previous approaches in terms of both RMSE and MAE metrics.

like conventional frame-based cameras, the event camera detects per-pixel brightness changes and outputs a stream of asynchronous events. This asynchronous nature brings several advantages, including low latency, high dynamic range, and high temporal resolution Gallego et al. (2020). Especially, event cameras Brandli et al. (2014); Rebecq et al. (2018); Liu et al. (2024) capture dynamic context and structural information, while frame-based cameras provide rich color and texture details. These two types of cameras complement each other effectively: event cameras excel in detecting rapid motion and changes in brightness with minimal latency, making them ideal for high-speed or low-light environments Falanga et al. (2020); Gallego et al. (2020). Meanwhile, frame-based cameras offer detailed, static scene representation. By integrating data from both, systems can achieve a more robust and comprehensive perception, which is especially advantageous in applications like autonomous driving, robotic vision, and augmented reality Gehrig & Scaramuzza (2024); Falanga et al. (2020); Chen et al. (2020); Gallego et al. (2020).

Current event-frame fusion methods often rely on concatenation Hu et al. (2020); Tomy et al. (2022) and attention mechanisms Munir et al. (2023); Sun et al. (2022); Zhou et al. (2023); Cao et al. (2021; 2024) to integrate event and frame data. While concatenating event- and frame-based features can provide modest performance gains, this approach fails to fully leverage the complementary characteristics inherent in each modality. Methods like those in Munir et al. (2023); Sun et al. (2022); Zhang et al. (2023); Cao et al. (2024) incorporate interactions between event-based and frame-based features, but they fall short of addressing the efficiency-performance trade-offs crucial for practical applications. Utilizing self-attention Munir et al. (2023) or cross-attention Sun et al. (2022); Zhang et al. (2023); Cao et al. (2024) in fusion modules can enhance performance, but structures are too complex.

To alleviate the limitations of existing fusion methods, we propose an effective multimodal hierarchical network for steering prediction, featuring a novel Energy-driven Cross-modality Fusion Module (ECFM). At the core of ECFM is an energy function that generates 3-D weights, enabling fusion by leveraging the strengths of each modality while introducing minimal additional learnable operations. Furthermore, we propose an energy-aware decoder to model the uncertainty, resulting in the decoder learning higher-quality predictive distributions, enhancing both safety and quality. As shown in Fig. 1, our approach outperforms other methods on two real-world datasets: DDD20 and DRFuser. Specifically, it achieves lower RMSE and MAE scores, indicating higher predictive accuracy in steering angle estimation.

In summary, our contributions are as follows:

- We introduce a novel energy-aware imitation learning framework for steering prediction by fusing events and frames effectively.

- The ECFM modules is proposed to enrich the extracted features with complementary information from both modalities, leading to improved prediction performance. In the ECFM, we employ an energy function to compute 3-D weights, which enhances model performance by adapting feature importance dynamically.

- We introduce an energy-aware decoder that incorporates a variance estimation branch to capture the uncertainty associated with steering predictions. To train this branch effectively, we employ an energy-aware loss, which encourages accurate modeling of predictive uncertainty.

- Our extensive experiments demonstrate that the proposed method outperforms SOTA approaches on both the DDD20 and DRFuser datasets. Our model achieves superior results compared to other popular methods, both qualified and quantified.

## 2 RELATED WORK

**Event-based Vision for Steering Prediction.** In the domain of event-based vision for steering prediction, the foundational work by Maqueda et al. Maqueda et al. (2018) introduced a two-channel event histogram to adapt convolutional architectures for processing event-based camera outputs. This approach leveraged transfer learning from pre-trained convolutional networks designed for image classification He et al. (2016), achieving strong performance for utilizing event-based data in steering prediction tasks. However, the dataset used in Maqueda et al. Maqueda et al. (2018) the DAVIS Driving Dataset 2017 (DDD17) Binas et al. (2017) is limited in terms of road diversity, weather, and daylight conditions, which constrains its applicability to varied driving scenarios. To address this gap, Hu et al. Hu et al. (2020) introduced DDD20, the longest end-to-end driving dataset to date, featuring a broader range of road types, weather, and lighting conditions. Additionally, Hu et al. proposed a deep learning approach to fuse event and frame data for predicting instantaneous human steering angles, advancing the potential of multimodal fusion in steering prediction. To formalize sensor fusion between an event camera and an RGB frame-based camera, the authors of Munir et al. (2023) proposed a convolutional encoder-decoder architecture called DRFuser. DRFuser incorporates self-attention mechanisms Dosovitskiy et al. (2021) to capture long-range dependencies between event-based and frame-based features, enhancing the fusion of these complementary modalities. To support this research, the authors collected a dataset featuring synchronized recordings from multiple sensors, including an RGB camera, an event camera, and Controller Area Network (CAN) data, under diverse weather and lighting conditions. This dataset spans urban, highway,

and suburban driving scenarios, providing a rich resource for event-frame fusion research in varied environments. Recently, Zhou et al. Zhou et al. (2024b) developed an end-to-end learning framework that fuses 2D LiDAR data with event data for steering prediction in autonomous racing. To facilitate this research, they created a multisensor dataset specifically tailored for steering prediction, encompassing synchronized 2D LiDAR and event data. Using this dataset, they established a benchmark and introduced a new fusion learning policy, advancing the exploration of multisensor fusion techniques.

**Various Fusion Modules for Events and Frames.** Multimodal fusion methods for combining event and frame data have been investigated across various vision tasks, including steering angle prediction Hu et al. (2020); Munir et al. (2023), deblurring Sun et al. (2022), semantic segmentation Zhang et al. (2023), and object detection Cao et al. (2021; 2024). These approaches leverage the complementary nature of event and frame-based data, enhancing performance across diverse applications by integrating the dynamic scene information from events with the rich spatial and texture details provided by frames. In Hu et al. (2020), an early fusion method is applied, where frames and event data are concatenated and fed into a convolutional network to predict the steering angle. Moreover, several middle fusion methods have been developed to enhance feature-level fusion between two modalities. For instance, in Cao et al. (2021), pixel-level spatial attention is used to amplify event-based features, thereby enhancing frame-based features and boosting performance. The cross-self-attention mechanism introduced in Sun et al. (2022) combines events and frames to improve deblurring results by aligning complementary features. In contrast, the authors of Munir et al. (2023) separately process events and frames with a self-attention module and then merge the outputs through summation to obtain fusion features. Additionally, the CMX Zhang et al. (2023) provides a unified approach for RGB-X semantic segmentation, utilizing tailored fusion strategies across different modalities. Recently, Cao et al. Cao et al. (2024) proposed a coarse-to-fine fusion strategy with the cross-modality adaptive feature refinement (CAFR) module to alleviate feature-modality imbalance and improve fusion quality.

## 3 PRELIMINARIES

In this section, we present the problem of steering angle prediction using both event-based and frame-based cameras. We describe the problem formulation and the event representation below.

### 3.1 PROBLEM FORMULATION

Imitation learning (IL), also known as learning from demonstrations, constitutes a machine learning paradigm wherein an agent assimilates the behavior demonstrated by an expert to master a task. The primary objective of IL lies in acquiring an agent model $\pi$ that aligns closely with an expert policy $\pi^*$. In this study, we employ the behavior cloning (BC) Bain & Sammut (1995) approach of IL to develop a lateral control policy responsible for mapping inputs to steering angles. Specifically, a dataset $D = (X^i, S^i)_i^T$ comprising $T$ instances is curated from the environment under the guidance of the expert policy $\pi^*$. Here, $X$ denotes high-dimensional observations encompassing event streams and frames, while $S$ represents the corresponding steering angle. The proposed model ($\pi$) is trained via supervised learning, utilizing the dataset $D$ in conjunction with a defined loss function $L$. The process can be formulated as follows:

$$\arg \min_{\pi} \mathbb{E}_{(X,S) \to D}[L(S, \pi(X))]. \tag{1}$$

where the Smooth L1 loss function and the proposed energy loss function serve as the objective function to gauge the disparity between the predicted steering angle $\pi(X)$ and the expert steering angle $S$.

### 3.2 EVENT REPRESENTATION

Event cameras capture changes in a scene as they happen, rather than recording frames at fixed intervals Gallego et al. (2020); Chen et al. (2020). The output from an event camera is a sparse and asynchronous stream of events. Each event $e_i$ can be represented as a tuple $e_i = (x_i, y_i, t_i, p_i)$, where $(x_i, y_i)$ denotes the pixel coordinates, $t_i$ is the timestamp, and $p_i \in \{+1, -1\}$ is the polarity,

Figure 2: The proposed model architecture consists of three main components: a dual-stream backbone network, ECFM modules, and an energy-aware decoder. The backbone network comprises two branches: the event-based ResNet at the bottom and the frame-based ResNet He et al. (2016) at the top. Each ECFM module operates to enhance features across different hierarchical scales. Subsequently, the energy-aware decoder is used to perform steering angle predictions.

indicating an increase or decrease in brightness, respectively. An event is triggered for each pixel $v_i = (x_i, y_i)^T$ when the change in brightness exceeds a threshold $C$, as expressed in Eq. 2:

$$E(v_i, t_i) - E(v_i, t_i - \Delta t_i) > p_i C, \quad p_i \in \{-1, +1\}. \tag{2}$$

where $E(v_i, t_i) \doteq log I(v_i, t_i)$ represents the brightness for each pixel $v_i$ at time $t_i$. This logarithmic transformation of the intensity $I(v_i, t_i)$ is often used to stabilize the response to brightness changes and improve the event camera's sensitivity to contrast variations. By recording events only when brightness change exceeds a threshold, the event camera efficiently captures meaningful scene dynamics while ignoring redundant information.

To enable event data processing with standard CNN-based architectures, we convert asynchronous events into grid-like representations. This conversion allows the sparse, asynchronous event streams to be structured in a spatial format compatible with convolutional layers. The process can be formulated as follows:

$$S_j^t = \mathbf{card}(e_i | T \cdot (j - 1) \le t_i \le T \cdot j). \tag{3}$$

where $S_j^t$ represents the $j$th event tensor of time interval $T$; $\mathbf{card}()$ is the cardinality of a set; $e_i$ is the $i$th event of the event stream.

## 4 METHOD

In this section, we provide a detailed explanation of our proposed model ($\pi$).

### 4.1 OVERVIEW

The proposed model ($\pi$), illustrated in Fig. 2, consists of a dual-stream backbone network, Energy-driven Cross-modality Fusion Modules (ECFM), and an energy-aware decoder. Both frames and event streams are processed through two parallel ResNet backbones, extracting features from each modality. To efficiently fuse these heterogeneous modalities, we introduce ECFM, which integrates frame-based and event-based features at each ResNet stage. The fused feature maps are subsequently passed to the energy-aware decoder, which outputs steering angle predictions. This framework enables effective leveraging of complementary visual information from both frame and event data.

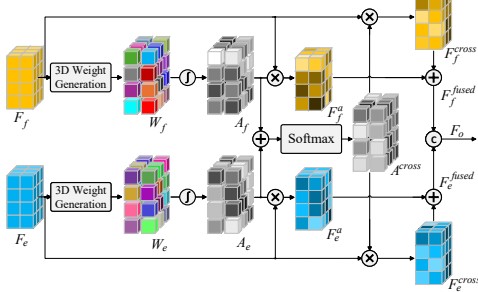

Figure 3: The architecture of the proposed ECFM. 3D weights are generated based on energy function.

## 4.2 Energy-driven Cross-modality Fusion Module

Frame-based features typically provide color, semantic, and texture information, while event-based features capture discriminative scene layout cues, making them complementary to frame-based features. To efficiently and effectively integrate these heterogeneous modalities, we propose a fusion strategy called the ECFM, illustrated in Fig. 3. This approach leverages the strengths of each modality almost without introducing additional learnable operations, enabling a seamless and effective fusion of both feature types.

Given two input features $F_f$ and $F_e$ produced by the convolutional block of the frame stream and event stream, respectively, we first generate 3-D weights by using energy function. Assuming that all pixels within a single channel follow the same distribution, it is reasonable to calculate the mean ($\mu$) and variance ($\sigma^2$) across all pixel features in that channel and reuse these values for each pixel feature within the channel Hariharan et al. (2012). This approach significantly reduces computational costs by avoiding the need to iteratively calculate $\mu$ and $\sigma^2$ for each position, making the process more efficient without sacrificing accuracy in feature normalization. The minimal energy $e_n$ can be formulated as follows:

$$e_n = \frac{4(\sigma^2 + \lambda)}{(n - \mu)^2 + 2\sigma^2 + 2\lambda}. \tag{4}$$

where $n$ represents the pixel feature and $\lambda$ denotes the coefficient factor. The mean ($\mu$) and variance ($\sigma^2$) are calculated as follows:

$$\mu = \frac{1}{M} \sum_{i=1}^{M} x_i, \sigma^2 = \frac{1}{M} \sum_{i=1}^{M} (x_i - \mu)^2. \tag{5}$$

where $M = H \times W$ represents the total number of pixels within the channel, with $H$ and $W$ denoting the height and width, respectively. To identify distinctive pixel features from their surrounding pixels, the energy function $e_n$ should be minimized. Consequently, the importance of each pixel feature can be represented by $\frac{1}{e_n}$, giving higher importance to pixel features with lower energy. The entire refinement process can be expressed as follows:

$$W_f = \frac{1}{\mathbf{E_f}}, W_e = \frac{1}{\mathbf{E}_e},$$
$$A_f = \mathrm{Sigmoid}(W_f), A_e = \mathrm{Sigmoid}(W_e). \tag{6}$$

where $\mathbf{E}$ groups all $e_n$ across channel and spatial dimensions. $\mathbf{E_f}$ and $\mathbf{E_e}$ represent energy functions with frame-based features $F_f$ and event-based features $F_e$ as inputs, respectively. $W_f$ and $W_e$ denote the generated 3-D weights. A Sigmoid function is applied to generate the activated vectors $A_f$ and $A_e$, effectively preventing excessively large values and maintaining stability.

The activated vectors are then applied on the input features ($F_f$ and $F_e$) in a multiplication manner. The procedure can be defined as:

$$F_f^a = A_f \otimes F_f, F_e^a = A_e \otimes F_e. \tag{7}$$

where $F_f^a$ and $F_e^a$ denote the enhanced frame-based features and event-based features, respectively. Additionally, the activated vectors $A_f$ and $A_e$ are integrated using a addition function to retain the valuable features from both the frame and event streams. This result is then passed through a **Softmax** function to normalize the output to a range between 0 and 1, producing the cross-modality activated vector $A^{cross}$. The process can be represented as follows:

$$A^{cross} = \mathrm{Softmax}(A_f + A_e). \tag{8}$$

Based on the $A^{cross}$, the fused features $F_f^{fused}$ and $F_e^{fused}$ are obtained by summing the enhanced features $F_f^a$ and $F_e^a$ with the $A^{cross}$-weighted features $F_f^{cross}$ and $F_e^{cross}$. This fusion process combines the strengths of both frame-based and event-based features, yielding a comprehensive

representation that leverages the distinct information from each modality. The process is described as follows:

$$F_f^{cross} = A^{cross} \otimes F_f, F_e^{cross} = A^{cross} \otimes F_e,$$
$$F_f^{fused} = F_f^{cross} + F_f^a, F_e^{fused} = F_e^{cross} + F_e^a. \tag{9}$$

The final output $F_o$ is obtained by applying a $\text{Conv}_{1\times1}$ layer to the concatenated fused features from the frame and event branches. Notably, only this part of the ECFM requires learnable parameters, which helps to minimize ECFM complexity while integrating the complementary information from both modalities. The computation process is defined as follows:

$$F_o = \text{Conv}_{1\times1}(\text{Concat}(F_f^{fused}, F_e^{fused})). \tag{10}$$

### 4.3 ENERGY-AWARE DECODER

We propose an energy-aware decoder for steering angle prediction. The features from four stages of the encoder are unified and integrated to get the balanced features. To enable reliable and safe predictions, we further introduce a variance estimation branch that captures the uncertainty associated with the steering outputs. An energy-aware loss is employed to train this branch, encouraging it to model predictive uncertainty effectively. By incorporating the energy-aware loss, our decoder learns higher-quality predictive distributions, enhancing both safety and quality.

**Energy-aware loss.** The energy loss is a strictly proper and non-local scoring rule Gneiting et al. (2008) used to evaluate probabilistic forecasts of multivariate quantities. Assuming that the steering prediction and variance estimation follow a multivariate Gaussian distribution, the energy loss can be expressed as:

$$L_E = \frac{1}{N} \sum_{n=1}^{N} (\frac{1}{M} \sum_{i=1}^{M} ||z_{n,i} - z_n|| - \frac{1}{2M^2} \sum_{i=1}^{M} \sum_{j=1}^{M} ||z_{n,i} - z_{n,j}||). \tag{11}$$

where $z_n$ denotes the ground-truth steering angle, and $z_{n,i}$ represents the $i^{\text{th}}$ sample from the $\mathcal{N}(\mu(x_n, \theta), \Sigma(x_n, \theta))$. The energy loss is defined based on the energy distance Rizzo & Székely (2016); Harakeh & Waslander (2021), which is a form of maximum mean discrepancy (MMD) Sejdinovic et al. (2013) that quantifies the distance between distributions of random vectors. MMD has been widely used for training generative models Li et al. (2015; 2017). In this work, we adopt the energy distance–a simple yet effective MMD–to estimate the divergence between two distributions from sample data. Given two independent random vectors $f, g \in \mathbb{R}^d$ with cumulative distribution functions $F$ and $G$, respectively, the squared energy distance is defined as:

$$D^2(F, G) = 2\mathbb{E}||f - g|| - \mathbb{E}||f - f'|| - \mathbb{E}||g - g'||. \tag{12}$$

where $f, f'$ are independent samples drawn from distribution $F$, and $g, g'$ are independent samples from distribution $G$. As shown in Rizzo & Székely (2016), the energy distance satisfies all the axioms of a metric, providing a rigorous measure for comparing distributions. In particular, it guarantees that $D^2(F, G) = 0$ if and only if $F = G$, thereby enabling assessment of distributional equality. Following Gneiting & Raftery (2007), Eq. 11 can be reformulated as:

$$L_E = \mathbb{E}||f - g|| - \frac{1}{2}\mathbb{E}||f - f'||. \tag{13}$$

It is straightforward to observe that Eq. 13 corresponds to the energy distance when only a single sample $g$ is available from distribution $G$. For multivariate Gaussian distributions, the energy score admits an efficient Monte Carlo approximation Gneiting et al. (2008), which can be expressed as:

$$L_E = \frac{1}{N} \sum_{n=1}^{N} (\frac{1}{M} \sum_{i=1}^{M} ||z_{n,i} - z_n|| - \frac{1}{2(M-1)} \sum_{i=1}^{M-1} ||z_{n,i} - z_{n,i+1}||). \tag{14}$$

Table 1: Comparison with SOTA methods on the DDD20 dataset. $\uparrow$ indicates metrics where higher values signify better performance, and $\downarrow$ indicates metrics where lower values are the better.

| Method | Model Type | RMSE ($\downarrow$) | MAE ($\downarrow$) |
|---|---|---|---|
| Bojarski et al. Bojarski (2016) | Frames only | 0.1574 | - |
| CNN-LSTM Xu et al. (2017) | | 0.1429 | - |
| Maqueda et al. Maqueda et al. (2018) | Events Only | 0.0716 | - |
| Hu et al. Hu et al. (2020) | Frames + Events | 0.0721 | - |
| FAGC Cao et al. (2021) | | 0.0501 | 0.0317 |
| EFNet Sun et al. (2022) | | 0.0430 | 0.0297 |
| DRFuser Munir et al. (2023) | | 0.0429 | 0.0291 |
| CMX Zhang et al. (2023) | | 0.0402 | 0.0273 |
| CAFR Cao et al. (2024) | | 0.0409 | 0.0275 |
| Ours | | **0.0381** | **0.0266** |

For each object instance in the minibatch, we draw a single set of $M$ samples from the distribution $\mathcal{N}(\mu(x_n, \theta), \Sigma(x_n, \theta))$. In our experiments, setting $M = 1000$ provides a balance between accuracy and efficiency, enabling the approximation in Eq. 14 to be computed with minimal computational overhead. The energy score is used to learn parametric distributions through differentiable sampling. Notably, due to its non-local nature, the energy distance encourages the model to assign probability mass close to the ground-truth target, even if not precisely at that value.

**Decoder structure.** The decoder employs an asymmetric design featuring three convolutional blocks, each composed of convolution, batch normalization, and ReLU activation layers. To bolster the model's generalization capability and mitigate overfitting, dropout layers are incorporated into the second and third convolutional blocks. Subsequently, after the convolutional blocks, two linear layers are utilized to predict the steering angle. The configuration of the first linear layer varies based on the input resolution. Please find more details in the **Appendix**.

## 5 EXPERIMENTS

In this section, we present an overview of the two public datasets used in this study, along with the evaluation metrics and implementation details. Following this, we conduct ablation studies to demonstrate the effectiveness of our approach. Finally, we evaluate the performance of our proposed method on both datasets.

### 5.1 COMPARISON WITH SOTA METHODS

We conduct a comprehensive evaluation by comparing our method with SOTA approaches on both the DDD20 and DRFuser datasets. The results highlight the strengths of our method, demonstrating superior performance in challenging scenarios. We present detailed comparisons and analyses of these results in the following:

**Experimental results on the DDD20 dataset.** Several works Bojarski (2016); Xu et al. (2017); Maqueda et al. (2018); Hu et al. (2020) have conducted experiments on the DDD20 dataset, providing important benchmarks in the field of steering prediction. In this study, we refer to the corresponding results reported in Munir et al. (2023). To provide a comprehensive comparison, we evaluate our method against SOTA alternatives in the Frame-Event domain, including FAGC Cao et al. (2021), EFNet Sun et al. (2022), DRFuser Munir et al. (2023), CMX Zhang et al. (2023), and CAFR Cao et al. (2024). The experimental results presented in Tab. 1 demonstrate that our method significantly outperforms other methods. Specifically, our method achieves lower RMSE and MAE scores, indicating higher accuracy.

**Experimental results on the DRFuser dataset.** Our results on the DRFuser dataset, as shown in Tab. 2, highlight that our method outperforms existing SOTA methods in the Frame-Event fusion domain. Specifically, the proposed method achieves substantial improvements in RMSE and

Table 2: Comparison with SOTA methods on the DRFuser dataset. ↑ indicates metrics where higher values signify better performance, and ↓ indicates metrics where lower values are the better.

| Method | Model Type | RMSE (↓) | MAE (↓) |
|--------|-----------|----------|---------|
| FAGC Cao et al. (2021) | | 0.1997 | 0.0240 |
| EFNet Sun et al. (2022) | | 0.2166 | 0.0241 |
| DRFuser Munir et al. (2023) | Frames + Events | 0.2919 | 0.0396 |
| CMX Zhang et al. (2023) | | 0.1971 | 0.0230 |
| CAFR Cao et al. (2024) | | 0.2209 | 0.0346 |
| Ours | | **0.0801** | **0.0102** |

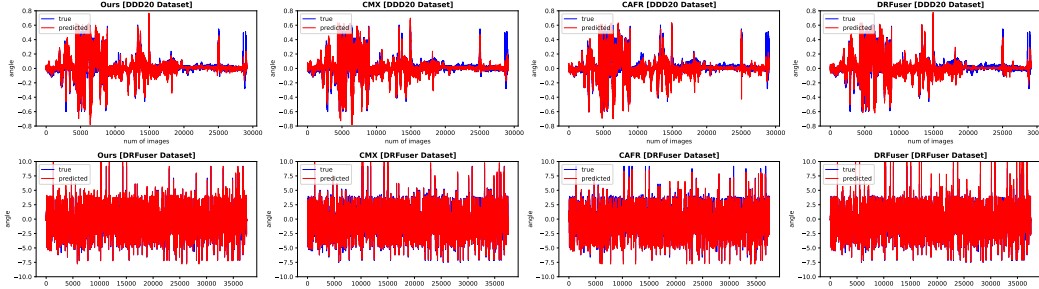

Figure 4: A quantitative comparison of our proposed method with CMX Zhang et al. (2023), CAFR Cao et al. (2024), and DRFuser Munir et al. (2023) on the DDD20 and DRFuser datasets.

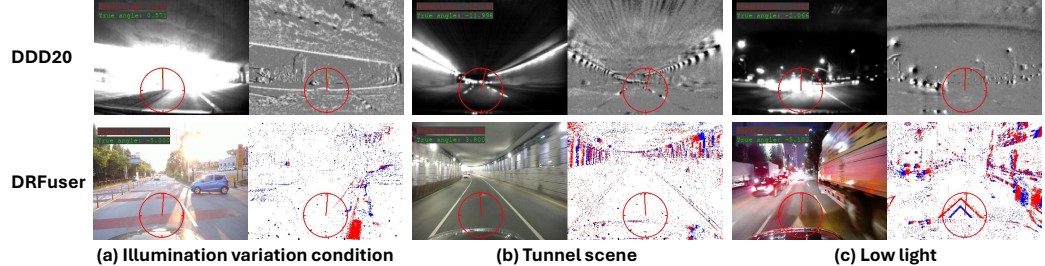

(a) Illumination variation condition     (b) Tunnel scene     (c) Low light

Figure 5: The visualization of steering angle prediction in comparison to ground truth on DDD20 (top) and DRFuser (bottom) datasets. Each pair of samples consists of frame data on the left and event data on the right.

MAE. Notably, compared to the second-best method, CMX Zhang et al. (2023), our approach yields significantly better RMSE (**0.0801 vs. 0.1971**) and MAE (**0.0102 vs. 0.0230**) scores. These results demonstrate the effectiveness of our method across a range of challenging driving scenarios, reaffirming its suitability for real-world deployment in diverse conditions.

**Results on steering predictions.** For a comprehensive evaluation of our proposed method, we conducted a quantitative comparison between our method and the leading methods: CMX Zhang et al. (2023), CAFR Cao et al. (2024), and DRFuser Munir et al. (2023). As depicted in Fig. 4, we present the steering prediction results of our method compared to other leading approaches on the DDD20 and DRFuser datasets. The visualized predictions clearly demonstrate the superior performance of our method, with predictions closely following the ground truth steering angles. Aside from a few outliers—likely caused by noise or inconsistencies in the test dataset collection—our method consistently achieves accurate and reliable steering angle predictions.

In Fig. 5, we present a visualization of the steering angle predictions produced by our model alongside the ground truth values for both the DDD20 (top) and DRFuser (bottom) datasets. Each sample pair consists of frame data (left) and corresponding event data (right), illustrating how our model effectively leverages the complementary information from both modalities. The selected scenarios encompass a range of real-world driving conditions, including varying lighting environments, diverse locations, and challenging scenarios such as tunnels, overexposed scenes, and crowded nighttime

settings. To ensure comprehensive evaluation, we tested our method across these varied conditions. The results demonstrate that our model reliably predicts steering angles with high precision, even under challenging lighting conditions and complex driving scenarios. Ground truth and predicted steering angles are marked on each sample pair in Fig. 5, highlighting our model's resilience in accurately capturing steering behavior in real-world driving environments. This consistency across both datasets underscores the robustness and adaptability of our approach, making it well-suited for practical autonomous driving applications, where environmental conditions are constantly changing.

## 5.2 ABLATION STUDY

To validate the effectiveness of the proposed module, we conducted ablation studies on the DRFuser dataset. The results are summarized in Tab. 3, demonstrating the impact of different fusion strategies on model performance.

**Multi-modal vs. single-modal.** Compared to single-modal approaches, even a simple fusion of events and frames using direct addition can improve performance, highlighting the complementary strengths of the two modalities. However, designing an efficient and effective fusion module to fully leverage these strengths remains an open challenge.

Table 3: Quantitative comparison with different ablation settings. "Add directly" (fusing events and frames by directly using addition operations); "Additive attention" (fusing events and frames by using basic attention operations; see the **Appendix** for more details).

| Method | RMSE ($\downarrow$) | MAE ($\downarrow$) |
|---|---|---|
| Events only | 0.5102 | 0.1501 |
| Frames only | 0.4609 | 0.1093 |
| Add directly | 0.3499 | 0.0434 |
| Additive attention | 0.2986 | 0.0403 |
| Ours | **0.0801** | **0.0102** |

**Effectiveness of our fusion strategy.** Exploring effective fusion methods for events and frames is essential. In Tab. 3, we present the results of two straightforward fusion strategies: "Add directly" and "Additive attention." While these methods highlight the complementary nature of frame-based and event-based features, they don't fully leverage this complementary information. To address this, we propose the ECFM, designed to integrate the complementary features from events and frames more effectively. Compared to "Add directly" and "Additive attention," our method significantly improves performance, achieving a reduction in RMSE and MAE (**0.0801 vs. 0.2986** and **0.0102 vs. 0.0403**, respectively). This demonstrates the superior capability of our method to harness information from both modalities for enhanced prediction accuracy.

**Influence of decoder design.** As demonstrated in Tab. 4, integrating features and applying energy loss both achieved better performance. Feature integration enables the model to gain more balanced features, thus increase the performance ( **0.0909 vs. 0.1016** and **0.0120 vs. 0.0122**), while the energy loss forces the model to be more accurate in outcomes, resulting in a more precise result (**0.0898 vs. 0.1016** and **0.0108 vs. 0.0122**). By combining the advantages of both component, we obtain the best result ( **0.0801** and **0.0102**).

Table 4: Quantitative comparison with different decoder design combinations.

| Integrate | Energy Loss | RMSE | MAE |
|---|---|---|---|
| ✗ | ✗ | 0.1016 | 0.0122 |
| ✓ | ✗ | 0.0909 | 0.0120 |
| ✗ | ✓ | 0.0898 | 0.0108 |
| ✓ | ✓ | **0.0801** | **0.0102** |

## 6 CONCLUSION

In this work, we introduce a novel energy-aware imitation learning for steering prediction using events and frames. Taking both frames and events as input, ECFM effectively harnesses the strengths of each modality almost without introducing additional learnable operations, enabling efficient and seamless fusion of their complementary features. Furthermore, an energy-aware decoder is proposed to capture the uncertainty associated with the steering outputs. Specifically, an energy-aware loss is used to guide the modeling of predictive uncertainty. Extensive experiments on two public real-world datasets, DDD20 and DRFuser, demonstrate that our approach outperforms current SOTA methods, achieving superior accuracy in terms of RMSE and MAE.

## ETHICS STATEMENT

This research adheres to the ethical standards of the ICLR community. All datasets used in our experiments are publicly available and contain no personally identifiable or sensitive information. Our models are developed solely for academic research purposes. We recognize that steering prediction techniques could potentially be applied in sensitive domains (e.g., autonomous driving, military), and we encourage their responsible use. We explicitly oppose harmful exploitation and strongly advocate for strict governance frameworks to ensure responsible development and deployment, minimizing potential societal risks.

## REPRODUCIBILITY STATEMENT

We will release the full code, configurations, preprocessing and evaluation scripts, and our trained weights upon acceptance.

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

## A   APPENDIX

### A.1   USE OF LLMs

We used LLMs only for grammar, wording, and formatting edits. All technical content, analyses, and reported results were authored and verified by the authors. There is no scientific claims or data that were generated by the LLMs.

### A.2   MORE EXPERIMENTAL SETTINGS

#### A.2.1   DATASETS

To evaluate the effectiveness of our proposed method, we conducted experiments on two public real-world datasets: DDD20 Hu et al. (2020) and DRFuser Munir et al. (2023). The details of each dataset are described as follows:

**DDD20.** The DDD20 dataset is an extensive driving dataset collected using a DAVIS camera, capturing approximately 4,000 km of urban and highway driving footage over a span of 51 hours. We follow the data selection and preprocessing method established by Hu et al. (2020). For event data, events are aggregated into 2D histograms with signed ON/OFF DVS event counts over 50 ms intervals. A total of 30 recordings were selected, representing a diverse range of road types and lighting conditions. To balance the dataset, frames with speeds below 15 km/h were excluded,

| Layer | Channel | Params | FLOPs | FPS | RMSE | MAE |
|-------|---------|--------|-------|-----|------|-----|
| L4 | 128 | 59.9M | 20.4 G | 49.13 | 0.0878 | 0.0115 |
| L4 | 256 | 64.1M | 24.2 G | 43.08 | 0.1094 | 0.0129 |
| L4 | 512 | 76.2M | 38.4 G | 44.68 | 0.0909 | 0.0120 |
| L3 | 128 | 59.9M | 20.4 G | 50.21 | 0.0940 | 0.0235 |
| L3 | 256 | 64.1M | 24.2 G | 48.79 | 0.0843 | 0.0119 |
| L3 | 512 | 76.2M | 38.4 G | 47.17 | **0.0801** | **0.0102** |

Table 5: Quantitative comparison across various integration layers and channel configurations.

and 70% of frames with steering angles between $\pm 5$ were pruned to mitigate the impact of skewed data distribution. Additionally, we removed outliers where steering angles exceeded three times the standard deviation in both the training and test sets. The training set contains 105,435 image pairs, while the test set includes 29,114.

**DRFuser.** The DRFuser dataset is a raw driving data collection used for end-to-end driving tasks, initially introduced in Munir et al. (2023). It includes 297.3 GB of data captured in an urban environment during both day and night. For our research, we selected 110 GB of data and performed preprocessing. Due to field conditions, some event data were empty because of the sensor settings; these data were removed from the dataset. The remaining data was then randomly split into training and test sets, ensuring no overlap. We also performed data synchronization to maintain consistency across event data, RGB frames, and vehicle control data (e.g., steering angle). The resulting dataset contains 71,274 paired samples of event and RGB image data with steering angle annotations, with 33,764 samples for training and 37,510 for testing.

### A.2.2 EVALUATION METRICS

For performance evaluation, we use two common metrics: Root Mean Square Error (RMSE) and Mean Absolute Error (MAE). The definitions of RMSE and MAE are formulated as follows:

$$
\begin{aligned}
RMSE &= \sqrt{\frac{1}{N}\sum_{i=1}^{N}(y_i - \hat{y}_i)^2}, \\
MAE &= \frac{1}{N}\sum_{i=1}^{N}|y_i - \hat{y}_i|.
\end{aligned}
\tag{15}
$$

where $y_i$ is the ground truth value, $\hat{y}_i$ is the predicted value, and $N$ is the total number of samples.

### A.2.3 IMPLEMENTATION DETAILS

The proposed network was implemented using PyTorch 1.8.0 Paszke et al. (2019) with CUDA 11.1 on an Ubuntu system, and training was conducted on an NVIDIA RTX 3090 GPU. We used a ResNet He et al. (2016) backbone pre-trained on the ImageNet-1K dataset Deng et al. (2009). To optimize training, we employed the AdamW Loshchilov & Hutter (2019) optimizer, an enhanced version of Adam that incorporates L2 regularization, improving training stability and effectiveness. The network was trained for a total of 100 epochs.

### A.3 MORE EXPERIMENTAL ANALYSIS

**Selection of integration layer and channel.** Tab. 5 presents a comparison of various integration settings. Notably, using layer 4 as the integration layer leads to suboptimal performance due to its low resolution and limited spatial detail. Although setting the channel dimension to 512 increases the number of parameters and FLOPs, it yields better performance while maintaining a respectable FPS.

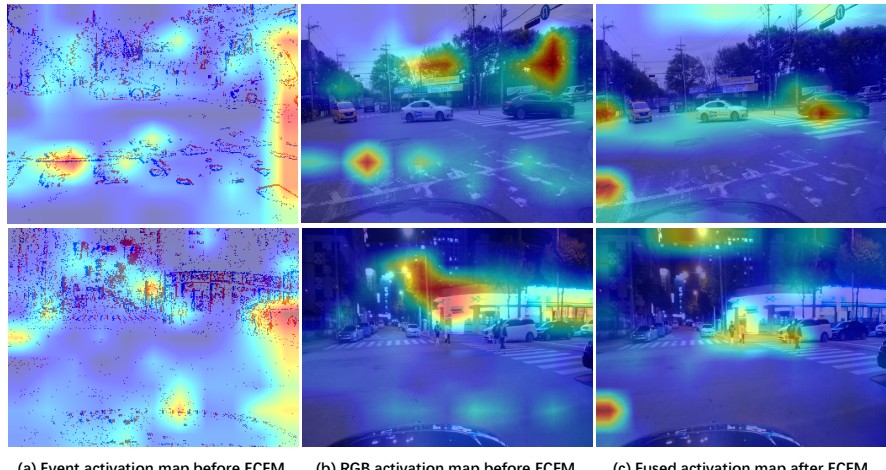

(a) Event activation map before ECFM    (b) RGB activation map before ECFM    (c) Fused activation map after ECFM

Figure 6: Representative examples of different activation maps on the DRFuser dataset are: (a) event activation map before ECFM; (b) RGB activation map before ECFM; and (c) fused activation map after ECFM.

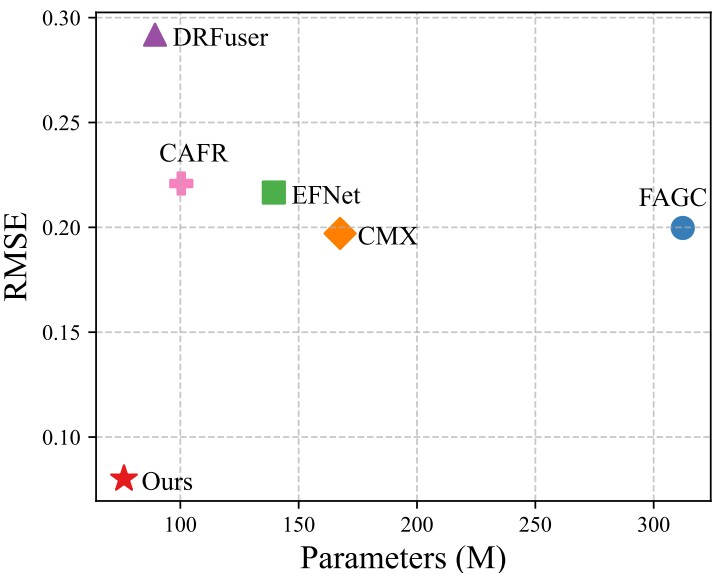

Figure 7: Performance vs. model size on the DRFuser dataset. Our proposed method achieves a new SOTA performance while utilizing significantly less number of parameters than previous methods.

**Visualization of activation maps.** In Fig. 6, we visualize the activation maps of the RGB and event modalities before and after applying the ECFM. These activation maps provide insights into how the model processes the input data. After the ECFM is applied, the fused activation map demonstrates an enhanced focus on significant regions, underscoring the module's effectiveness in extracting and integrating salient features from both modalities. By combining the complementary strengths of event and RGB features, the ECFM produces more refined and focused activations, leveraging the synergy between the two modalities for improved feature representation and model performance.

**Performance vs. model size.** As shown in Fig. 7, our proposed method achieves a lower RMSE while utilizing fewer model parameters. These results highlight the method's strong balance between accuracy and efficiency, making it well-suited for real-world applications.

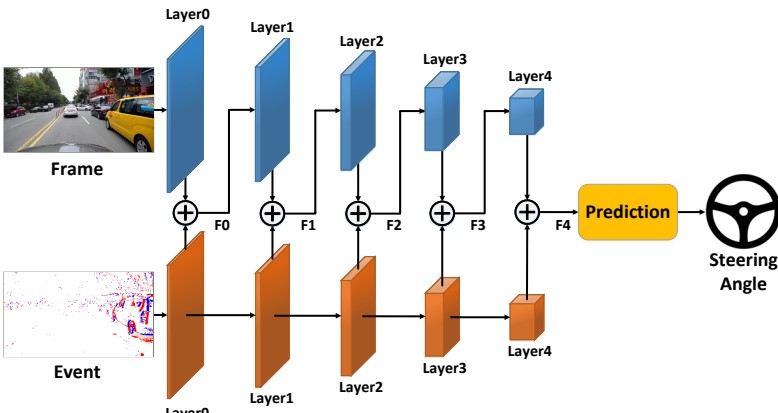

Figure 8: The addition baseline model consists of three main components: a dual-stream backbone network, addition modules, and a prediction decoder.

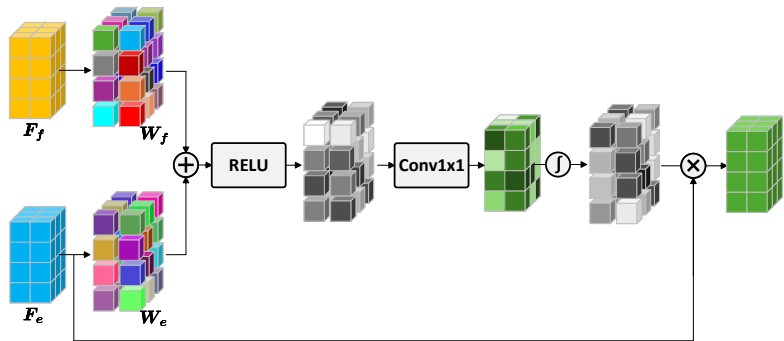

Figure 9: The architecture of the additive attention.

### A.4 DETAILS FOR BASELINE MODELS

To demonstrate the efficacy of the proposed ECFM fusion module, we conducted a comparative analysis with baseline fusion methods. To ensure a fair comparison, we maintained the same experimental settings as in our proposed method, with the only difference being the replacement of the fusion module with baseline methods, specifically direct addition and additive attention Bahdanau (2014). Through these experiments, we aim to highlight the superiority of our ECFM fusion module in enhancing the performance of the event-frame fusion model, demonstrating its ability to improve accuracy and robustness compared to the baseline methods. Further details about these two baseline models used in the ablation study are provided in the following.

**Fusion module using direct addition.** The direct addition approach does not employ any attention mechanisms. Instead, it processes the two modality features by directly adding them, treating all features equally without prioritizing their significance. This straightforward fusion process is illustrated in Fig. 8. The fused result at each stage can be mathematically expressed as:

$$F_i = F_f + F_e, \quad i \in \{0, 1, 2, 3, 4\}. \tag{16}$$

where $F_f$ and $F_e$ represent the feature maps extracted from the frame and event modalities, respectively.

**Fusion module using additive attention.** In contrast, the additive attention module selectively attends to the important parts of the input by computing a weight for each feature, enabling the model to focus on the most relevant features. The working principle of this mechanism is depicted in Fig. 9. Initially, the two generated weight matrices ($W_f$ for frame features and $W_e$ for event features) are

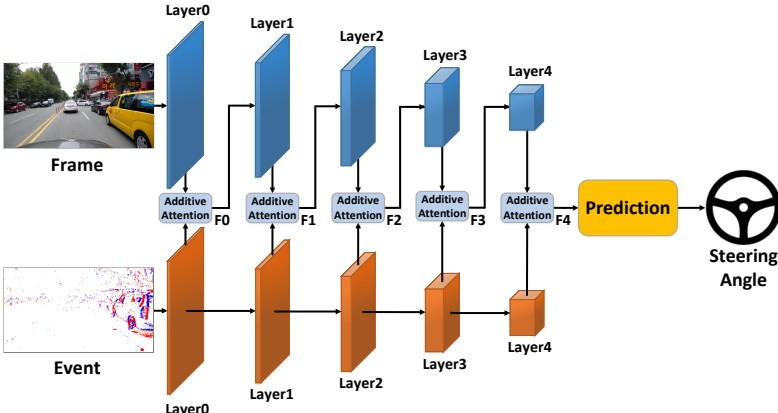

Figure 10: The additive attention baseline model consists of three main components: a dual-stream backbone network, additive attention modules, and a prediction decoder.

Table 6: The detailed architecture of the decoder in the proposed network.

| Architecture | Layers | Input Dimension | Output Dimension |
|---|---|---|---|
| | Convolution 2D | 2048 | 1024 |
| | Batch normalization | 1024 | 1024 |
| | ReLU | 1024 | 1024 |
| | Convolution 2D | 1024 | 512 |
| | Batch normalization | 512 | 512 |
| | Dropout | 512 | 512 |
| | ReLU | 512 | 512 |
| Decoder with ResNet-50 | Convolution 2D | 512 | 256 |
| | Batch normalization | 256 | 256 |
| | Dropout | 256 | 256 |
| | ReLU | 256 | 256 |
| | Linear layer | 256*H*W | 512 |
| | ReLU | 512 | 512 |
| | Linear layer | 512 | 1 |

added together. Activation functions (ReLU and Sigmoid) and a $\mathrm{Conv}_{1\times1}$ layer are then applied to produce the attention coefficients, denoted as $\alpha_i$. These coefficients $\alpha_i$ identify the salient regions in the input, which are subsequently fused with the event data through element-wise multiplication. The complete framework is illustrated in Fig. 10, and the process can be mathematically described as follows:

$$
\begin{aligned}
W_f &= \mathrm{BN}(\mathrm{Conv}_{1\times1}(F_f)), \\
W_e &= \mathrm{BN}(\mathrm{Conv}_{1\times1}(F_e)), \\
\alpha_i &= \mathrm{Sigmoid}(\mathrm{BN}(\mathrm{Conv}_{1\times1}(\mathrm{ReLU}(W_f + W_e)))), \\
F_i &= F_e \cdot \alpha_i, \quad i \in \{0, 1, 2, 3, 4\}.
\end{aligned}
\tag{17}
$$

where BN denotes batch normalization, and $\cdot$ represents element-wise multiplication. This mechanism effectively enhances the fusion process by emphasizing salient features while suppressing irrelevant ones.

### A.5 More Model Details

**Dual-stream backbone.** The backbone consists of two branches: event-based and frame-based ResNets He et al. (2016). Each ResNet comprises five layers, labeled as $\{\mathrm{Layer}_0, \mathrm{Layer}_1, \mathrm{Layer}_2, \mathrm{Layer}_3, \mathrm{Layer}_4\}$. The resolution of the feature maps gradually decreases

from $L_0$ to $L_4$, while the features' resolution remains constant at each individual layer. To extract semantically richer and more meaningful features, residual learning is utilized. In this study, a ECFM module is strategically placed between the two layers to further enhance the learning process.

**Decoder structure.**  To enable the prediction of steering angles from extracted features, a lightweight yet effective decoder was devised. This decoder network is designed to maintain computational efficiency while achieving high predictive accuracy. The decoder consists of the following components:

- *Cnvolutional blocks*: Each block comprises a convolutional layer for feature extraction, followed by batch normalization to stabilize and accelerate training, and a ReLU activation function to introduce non-linearity.
- *Dropout layers*: Incorporated in the second and third convolutional blocks to prevent overfitting and improve generalization.
- *Linear layers*: Two fully connected layers are added after the convolutional blocks to refine features and predict the steering angle.

Tab. 6 presents the detailed architecture of the decoder within the proposed framework. The decoder with the ResNet-50 encoder consists of three convolutional blocks. The first block uses 2D convolution layers, taking input from a 2048-dimensional feature map and producing a 1024-dimensional feature map. Batch normalization and ReLU activation layers follow each convolution. The second block processes the output from the first, applying a 2D convolution to produce a 512-dimensional feature map, followed by batch normalization and dropout layers. The third block takes the output from the second and applies a 2D convolution to generate a 256-dimensional feature map, followed by batch normalization and dropout layers. Finally, two fully connected layers are employed to predict the steering angle. The first linear layer takes the output from the third block as input, and the second linear layer processes the output from the first linear layer.

## A.6  LIMITATION

Our evaluations are limited to this specific task. It raises an open question about the generalizability of our method across other key perception tasks, such as semantic segmentation, depth prediction, and object detection. Extending our method's application to these areas could offer valuable insights into its adaptability and effectiveness in handling a broader range of challenges.

