# OpenReview forum: "Energy-Aware Imitation Learning for Steering Prediction Using Events and Frames"
_ICLR.cc/2026/Conference — ICLR 2026 Conference Withdrawn Submission_

### Official Review · Reviewer_eKq3 · 2025-10-26

**Soundness:** 3
**Presentation:** 3
**Contribution:** 3
**Rating:** 6
**Confidence:** 2

**Summary:**

This paper proposes a new method for predicting steering angles in self-driving cars by combining regular camera frames and data from event cameras. The main ideas are a new fusion module called ECFM, which cleverly mixes the two types of data without adding many parameters, and a special decoder that can also estimate how uncertain its own prediction is. The experiments show that this method works much better than previous ones on two real-world driving datasets.

**Strengths:**

1. The method achieves state-of-the-art performance on both datasets, significantly outperforming previous approaches.

2. The Energy-driven Cross-modality Fusion Module (ECFM) seems like a very clever and efficient way to combine frame and event data. The idea of using an "energy function" to do this with almost no extra learnable parameters is a neat contribution.

3. I appreciate that the authors didn't just focus on accuracy. The energy-aware decoder, which predicts uncertainty, is a great feature for safety-critical applications like autonomous driving. It's important for a system to know when it's not confident.

**Weaknesses:**

1. The core concept of the "energy function" (Eq. 4) a bit hard to grasp intuitively. While the math is provided, a simpler, more high-level explanation of why this particular formula works to measure feature importance would improve the paper's clarity for a broader audience.

2. The paper emphasizes that its fusion module is efficient, but the main comparison tables (Tables 1 and 2) are missing a direct comparison of inference speed (e.g., in Frames Per Second, FPS) against the other state-of-the-art methods.

**Questions:**

1. Could you please provide a more intuitive, high-level explanation of the "energy function" in Equation (4)?

2. Could you provide a comparison of the inference speed (e.g., in FPS) of your model against the other SOTA methods?

---

### Official Review · Reviewer_qVud · 2025-10-27

**Soundness:** 2
**Presentation:** 1
**Contribution:** 2
**Rating:** 2
**Confidence:** 3

**Summary:**

This is an experimental paper that proposes a new approach for steering angle prediction from events and frames camera inputs. It incorporates specific “energy-aware” architectures for fusion and decoder modules, as well as an energy distance used as a loss, enabling a form of uncertainty prediction. The model is then evaluated on two popular benchmarks in the field, showing competitive RMSE and MAE performance.

**Strengths:**

The method seems to provide competitive RMSE and MAE performance compared to numerous competitors in the presented benchmarks. Several ablations are performed, including (i) the importance of the multimodal compared to frame-only inputs, (ii) the role of some components of the fusion strategy, and (iii) the influence of the decoder design (with and without feature integration and energy loss). This justifies the approach to some extent.

**Weaknesses:**

The paper is hard to read and not very clear.

The paper insists on the importance of the energy loss to perform probabilistic forecasts. However, the ability of the method to perform uncertainty quantification is not demonstrated. For each example $n$, did you use the mean of the predicted distribution to compute the RMSE and MAE ? Did you try to interpret / quantify the relevance of the predicted distribution from the model ? In ambiguous cases, such as in an intersection, is the model able to predict multiple “modes” for the steering angles ? Calibration could be evaluated as in [A].

The discussion in L303-L323 should not be in the main paper; it is way too long because these are widely known elements,and there is no new contribution on that account.

The appendix doesn’t contain enough experimental details. In the next revision, all the used hyperparameters (e.g, weight decay, learning rate, batch size, etc.) should be detailed in the Appendix.

**Typos/issues noticed:**
Sentence flow issue in L96-97.

Eq (1): $(X,S) \sim D$ rather?

Eq (2): Shouldn't it be the opposite inequality when $p_i = -1$ ? If I am not mistaken, it should rather be $| E\left(v_i, t_i\right)-E\left(v_i, t_i-\Delta t_i\right) | >C$

The role of Eq. (3) is not clear.

Where does the expression (4) come from?

We don't see the green/red text well in Figure 5.

**References**

[A] Hubschneider, Christian, Robin Hutmacher, and J. Marius Zöllner. "Calibrating uncertainty models for steering angle estimation." In 2019 IEEE intelligent transportation systems conference (ITSC), pp. 1511-1518. IEEE, 2019.

**Questions:**

See weaknesses above, also:

* Can you verify experimentally the relevance of the distributional outputs (apart from the mean) ? Why does energy loss enhance accuracy ? I would rather expect a better distribution match, which doesn’t necessarily imply better accuracy as measured by RMSE and MAE.

* In Table 4, what is the loss when energy loss is not used?

* The method seems to be strong in terms of RMSE and MAE, but there is no real explanation nor interpretation for the gain in performance. For instance, why is the proposed fusion operation more effective than the baselines in Table 3 ?

---

### Official Review · Reviewer_W8ZD · 2025-11-01

**Soundness:** 2
**Presentation:** 2
**Contribution:** 2
**Rating:** 2
**Confidence:** 3

**Summary:**

This paper addresses the problem of robust perception for steering angle prediction in autonomous driving, where standard frame-based cameras fail in challenging high-speed motion or adverse lighting conditions. The authors propose an energy-aware imitation learning framework that fuses information from standard frames with data from event cameras, which are robust to such issues. The framework consists of a dual-stream (event and frame) backbone and advances 2 contributions. (1) an Energy-driven Cross-modality Fusion Module (ECFM) that integrates features from the two modalities without learning, by essentially leveraging per-channel statistics in an energy formulation to generate 3D fusion weights.
(2) an energy-aware decoder equipped with a variance prediction branch for the steering angle uncertainty. To train this decoder the authors propose an energy-aware loss, here a proper scoring rule estimating the energy distance (in fact MMD) between the predicted distribution and the target label.

The approach is trained with imitation learning (behavior cloning) for steering angle prediction. The experiments on two public datasets, DDD20 and DRFuser, show that the proposed method achieves state-of-the-art performance, outperforming previous fusion methods in terms of RMSE and MAE.

**Strengths:**

**Significance**
- This work tackles a critical problem for real-world deployment of autonomous systems: robust perception in challenging (e.g., low-light, high-speed, over-exposure) scenarios where standard cameras fail.
- Such cameras could be indeed interesting for autonomous driving


**Originality**
- The proposed approach for repurposing the energy-distance looks novel to me for this setting.


**Quality**
- The reported performance is strong compared to the existing baselines on two public datasets commonly used in event-cameras literature
- The approach displays good per-parameter efficiency by achieving high performance with few parameters


**Clarity**
- The authors report several qualitative visualizations to show the behavior of their approach in successful cases (steering predictions, activation maps, overall gap w.r.t. ground-truth labels). However there are no visualizations showing how the method can fail.
- Implementation information of the used hyper-parameters related to architecture is detailed

**Weaknesses:**

**Confusing terminology**
- The paper overloads the term "energy" to describe two different and unrelated concepts causing quite a confusion across the paper.
- In the fusion part, ECFM, the energy is a form of heuristic for a non-learnable function that uses channel mean and variances to generate fusion weights.
- In the energy decoder, "energy" refers to the energy distance, based on a proper scoring rule from probabilistic forecasting to measure the divergence between two distributions
- The ECFM part could be called instant something like "feature statistics-based fusion" or something like that to improve clarity.
- In addition the term $e$ used to denote the energy for ECFM is also used for event camera events.

**Limited scope**
- The approach argue that event cameras are highly useful for autonomous driving, however the proposed setup of doing imitation learning on steering angle prediction is quite different from dominant practices in autonomous driving.
- In autonomous driving, modern imitation learning methods (open-loop) typically predict several waypoints and not just a single volatile action, to ensure more stable intermediate representations which are easier to follow [a], [b], [c]
- Action prediction (steering angle and throttle prediction) are often used in closed-loop settings where models are trained with reinforcement learning, but even there the trend is to go towards prediction trajectories. Besides, the current approach predicts just the steering angle.
- In addition, my understanding is that the setup is quite limited as it does not consider temporal information, just pairs of frame and event cameras, while there are no low-speed maneuvers, below 15 km/h (line 647)
- These are limitations of the existing datasets, but unfortunately they don't serve the endeavor of the authors. I would suggest trying simulated data, e.g., CARLA, to show actual driving setup. Alternatively, focusing on perception-only tasks, e.g., object detection, could be interesting as event-camera detections would be definitely useful to assist typical camera (+lidar) setups.

**Limited validation of the contributions**
- The authors claim that their energy-aware decoder captures the uncertainty to learn higher-quality predictive distributions and enhance both safety and quality.
- However the experiments report still only point-prediction accuracy (RMSE, MAE) and there is no evaluation of the quality of the predicted uncertainty
- This can be validated by measuring and analyzing the negative log-likelihood (NLL), or threshold predictions based on uncertainty and measure accuracy scores.
- Additionally perturbations can be added to the inputs to simulate sensor corruption or distribution shift, in the style of Robo-BEV, ImageNet-C, to see how uncertainty helps there [d], [e]
- The hyper-parameter $\lambda$ introduced in the energy formulation (eq. 4) appears crucial and its value, selection process and sensitivity are not discussed.
- Also why this energy distance was used and not something else like Gaussian-likelihood using predicted variance [f]
- If I understand correctly, the DRFuser dataset is a custom variant of the original dataset by Munir et al. select by the authors. The authors mention also the use of a random training and test split, but no validation set is mentioned. How are results validated?



**Clarity and motivation**
- The proposed energy function for fusion is simply dropped in eq.4 but there is no description on where does it come from, how did the authors reach it and what it is expected to do.
- In general I thing that the ECFM's motivation is quite weak.
    + Why such a specific, per-dataset tuned and complex heuristic is needed?
    + Why this hand-crafted approach should be better than a simple learnable attention mechanism? The ablation in Table 3 only compares against additive attention and not against a more modern and stronger baseline like transformer-based cross- or self-attention
- In general the fusion part has very limited ablations on the impact of the chosen components


**Minor Misc.**
- At some point in the paper the authors shift from "energy" to describe the fusion to "integration"
- With natbib please use $\citep$ accordingly for the references
- The related work section on event-based vision for steering prediction mixes methods and datasets and is not easy to follow
- Self-attention was not invented by Dosovitskiy et al.


**References:**

[a] Bansal et al., ChauffeurNet: Learning to Drive by Imitating the Best and Synthesizing the Worst, arXiv 2018

[b] Chitta et al., TransFuser: Imitation with Transformer-Based Sensor Fusion for Autonomous Driving, TPAMI 2023

[c] Hu et al., lanning-oriented autonomous driving, CVPR 2023

[d] Xie et al., Benchmarking and Improving Bird's Eye View Perception Robustness in Autonomous Driving, TPAMI 2025

[e] Hendrycks & Dietterich, Benchmarking Neural Network Robustness to Common Corruptions and Perturbations, ICLR 2019

[f] Kendall & Gal, What Uncertainties Do We Need in Bayesian Deep Learning for Computer Vision?, NeurIPS 2017

**Questions:**

This paper takes an interesting direction of study, but the validation are currently not that conclusive and much of the motivation is not well described.

I find the endeavor of the authors nice. However I do have several concerns regarding the limited scope of the experiments, the limited ablations, weak missing motivation and the limited scope application of the reactive steering imitation for autonomous driving.

My current rating is leaning towards reject at this time, but I'm looking forward for the rebuttal.

Here are a few questions and suggestions that could be potentially addressed in the rebuttal or in future versions of this work (please note that suggested experiments are not necessarily expected to be conducted for the rebuttal):

1. Please clarify the confusing use of the term "energy" for the two different concepts (ECFM heuristic and energy score).

2. Detail and explain the formulation of the energy function in equation (4). What is the value of the hyper-parameter $\lambda$? How was it chosen and how sensitive is the model performance to this value?

3. A core contribution is the energy-aware decoder for uncertainty. The experiments, however, only report RMSE/MAE. Can you provide any quantitative evaluation of the quality of the predicted uncertainty (e.g., NLL, performance w.r.t. different uncertainty thresholds) to validate the claims about safety and quality.

4. The ECFM is shown to be superior to additive attention. How does it compare in both performance and efficiency to a more standard baseline like a transformer-based cross-attention module?

5. Have the authors tested the ECFM fusion on other perception tasks for event cameras, e.g., object detection?

---

### Official Review · Reviewer_6t3M · 2025-11-01

**Soundness:** 2
**Presentation:** 1
**Contribution:** 2
**Rating:** 2
**Confidence:** 3

**Summary:**

This work considers the problem of steering-angle prediction using multi-modal inputs
containing both frame-based camera images and event-based ones. Unlike the traditional
frame-based camera, the event-based cameras capture pixels whose lightings change, and
thus can capture movements better, but pose a challenge in fusing features with frame-based
ones. This paper proposes a novel fusion method, called Energy-driven Cross-modality Fusion
Modules (ECFM), where per-pixel features from different modalities are weighted sum together
based on a heuristic energy function. The energy function leads to higher weights on features
that are more standard-deviations from the mean of the features.

Experiments are conducted on the DDD20 and DRFuser driving datasets, which contain
event-based camera images. The proposed method achieved the SOTA RMSE and MAE for
steering angle prediction.

**Strengths:**

- Originality: This work proposed a modality fusion module for two types of camera image
features. The module generates weights on image features based on feature populate
statistics, instead of through typical stochastic gradient descents.
- Significance: The proposed architecture achieved the SOTA performance.

**Weaknesses:**

- Clarity: The presentation has several places to be improved. Here are some examples:
- The paper title contains the word “energy”, which can be an overloaded term.
However, readers do not realize it is related to energy distance of distributions until
about line 300, well into method sections, and after “energy” is mentioned about 20
times. Hence, an early “preliminary” section to define these terms could be helpful.
- Eq 4 defines a “minimal energy” without defining what it minimizes.
- It is not clear what motivates the use of “energy distance” for loss, or the use of
inverse-min-energy for weighting pixel features.
- In the experiment section, there is no introduction to the dataset, metrics and
baselines, which makes it difficult to evaluate this work.
- The title mentions “imitation-learning”, but the method really is just supervised
learning for steering angles. Evaluation also doesn’t include any “on-policy”
evaluation which IL typically involves.

**Questions:**

- In Eq 4. How is the energy defined? What motivates energy-based weightings on the
per-pixel features? From the math, it seems to give higher weights to pixels that are
farther from the means.
- It is unclear what the purpose of the parts of the fusion module are, e.g., why the
“cross” weight is needed? There is also no ablation on this. It may be interesting to
show the process to arrive at the proposed structure.

---

### Note · Authors · 2025-11-13

I have read and agree with the venue's withdrawal policy on behalf of myself and my co-authors.